# Live imaging reveals the cellular events downstream of SARM1 activation

Kwang Woo Ko[1], Laura Devault[1], Yo Sasaki[2], Jeffrey Milbrandt[3]*, Aaron DiAntonio[4]*

[1]Washington University School of Medicine, St Louis, United States; [2]Genetics, Washington University School of Medicine, St Louis, United States; [3]Genetics, Hope Center for Neurological Disorders, Washington University School of Medicine, St Louis, United States; [4]Developmental Biology, Needleman Center for Neurometabolism and Axonal Therapeutics, Washington University School of Medicine, St Louis, United States

**Abstract** SARM1 is an inducible NAD$^+$ hydrolase that triggers axon loss and neuronal cell death in the injured and diseased nervous system. While SARM1 activation and enzyme function are well defined, the cellular events downstream of SARM1 activity but prior to axonal demise are much less well understood. Defects in calcium, mitochondria, ATP, and membrane homeostasis occur in injured axons, but the relationships among these events have been difficult to disentangle because prior studies analyzed large collections of axons in which cellular events occur asynchronously. Here, we used live imaging of mouse sensory neurons with single axon resolution to investigate the cellular events downstream of SARM1 activity. Our studies support a model in which SARM1 NADase activity leads to an ordered sequence of events from loss of cellular ATP, to defects in mitochondrial movement and depolarization, followed by calcium influx, externalization of phosphatidylserine, and loss of membrane permeability prior to catastrophic axonal self-destruction.

*For correspondence:
jmilbrandt@wustl.edu (JM);
diantonio@wustl.edu (ADiA)

## Editor's evaluation

The authors have responded to all essential criticisms, supporting the elucidation of a sequence of cellular events that reside downstream of SARM1 activation.

## Introduction

SARM1 is the central executioner of pathological axon degeneration, an early feature of many neurodegenerative diseases (*Figley and DiAntonio, 2020*; *Krauss et al., 2020*). SARM1 is the founding member of the TIR-domain family of NAD$^+$ hydrolases (*Essuman et al., 2018*; *Essuman et al., 2017*), and a metabolic sensor activated by disrupted NAD$^+$ homeostasis (*Figley et al., 2021*; *Gilley et al., 2015*; *Sasaki et al., 2016*). Activation of the SARM1 NADase depletes cellular NAD$^+$ and initiates a metabolic crisis that ultimately leads to axon degeneration and/or neuronal cell death (*Essuman et al., 2017*; *Gerdts et al., 2015*). SARM1 is a compelling target for therapeutic intervention, as loss of SARM1 is profoundly protective in animal models of multiple neurodegenerative diseases including nerve injury, peripheral neuropathies, traumatic brain injury, glaucoma, retinitis pigmentosa, and Leber congenital amaurosis (*Geisler et al., 2016*; *Gerdts et al., 2013*; *Henninger et al., 2016*; *Ko et al., 2020*; *Osterloh et al., 2012*; *Ozaki et al., 2020*; *Sasaki et al., 2020b*; *Turkiew et al., 2017*). Moreover, as an enzyme, SARM1 is a druggable target, and both small molecule inhibitors and gene therapeutics effectively block axon degeneration (*Bosanac et al., 2021*; *Geisler et al., 2019*; *Hughes et al., 2021*). Recently, there has been tremendous progress in dissecting the structure of SARM1

(*Bratkowski et al., 2020*; *Jiang et al., 2020*; *Sporny et al., 2020*), the mechanism by which SARM1 is autoinhibited in healthy neurons (*Shen et al., 2021*) and activated in diseased neurons (*Figley et al., 2021*), and its role as an NAD$^+$ hydrolase (*Essuman et al., 2017*; *Horsefield et al., 2019*; *Zhao et al., 2019*). However, the events downstream of NAD$^+$ loss but prior to catastrophic axon fragmentation are much less well understood.

SARM1 and its NADase activity are essential for injury-induced axon degeneration, and the SARM1 enzyme is activated within 1–2 hr after injury in cultured DRG neurons (*Sasaki et al., 2020a*). Axonal fragmentation occurs much later in this system, with axons fragmenting approximately 4–6 hr after injury. Numerous molecular and cellular events occur in the time between SARM1 activation and axon loss, including calcium influx, mitochondrial stalling and depolarization, loss of ATP, and disrupted membrane integrity. We reasoned that temporally ordering these events would give insights into causal relationships among these degenerative mechanisms. Unfortunately, studies in bulk culture are not appropriate for assessing the temporal sequence because axon loss is asynchronous, and so this effort requires live imaging. Prior live imaging studies have demonstrated that hours after injury there is a large influx of calcium that precedes axonal fragmentation, and blocking this late calcium entry with EGTA inhibits axon fragmentation (*Adalbert et al., 2012*; *Loreto et al., 2015*; *Vargas et al., 2015*; *Wang et al., 2000*; *Yong et al., 2020*). However, no comprehensive live imaging analysis of the cellular and molecular events underlying axon degeneration has been reported.

Here, we explore the cellular events that occur in injured axons following SARM1 activation. First, we investigate whether changes to mitochondria and calcium require SARM1 NADase activity, as prior studies used a complete knockout and so left open the possibility of NADase-independent functions. Indeed, such an NADase-independent function was recently described in *Drosophila* for organelle stalling after injury (*Hsu et al., 2021*). Next, we revisit the role of both intracellular and extracellular calcium in axon degeneration. We confirm that blocking extracellular calcium influx blocks axon fragmentation (*Vargas et al., 2015*; *Villegas et al., 2014*; *Wang et al., 2000*; *Witte et al., 2019*); however, these apparently morphologically intact axons are not metabolically active, as mitochondria are immobile and depolarized. We then develop a live imaging approach in cultured DRG neurons with single axon resolution and assess dynamic changes to calcium, mitochondria, ATP, and the plasma membrane. Our findings describe an ordered series of events in which (1) ATP is lost, (2) mitochondria stop moving and subsequently depolarize, (3) extracellular calcium enters the axons, (4) phosphatidylserine is exposed on the outer leaflet of the plasma membrane, and (5) the membrane loses integrity. This work identifies a stereotyped cascade of dysfunction following SARM1 activation, and highlights ATP loss as the likely key intermediate between NAD$^+$ cleavage and widespread dysfunction in injured axons.

## Results

### SARM1 NADase activity promotes mitochondrial stalling and calcium influx in injured axons

SARM1 is an injury-activated NAD$^+$ hydrolase, and this enzymatic activity is required for injury-induced axon degeneration (*Essuman et al., 2017*). However, it is unclear whether all SARM1-dependent processes require this enzymatic function. Recently, Hsu et al. working in *Drosophila* demonstrated that injury-dependent organelle stalling can be SARM1-dependent but NADase-independent (*Hsu et al., 2021*). Previously, Loreto et al presented the even more surprising finding that injury-dependent mitochondrial stalling in superior cervical ganglion axons did not depend on SARM1, although loss of mitochondrial potential did depend on SARM1 (*Loreto et al., 2015*). Here, we test the applicability of these findings to mammalian sensory neurons, assessing mitochondrial movement in cultured mouse dorsal root ganglion (DRG) neurons and assaying the requirement for SARM1 NADase activity. This is of particular interest because SARM1 is a mitochondrial associated protein and so alternate mechanisms of action are plausible. We cultured DRG neurons from SARM1 knockout (KO) embryos and used lentivirus to express either GFP, SARM1, or catalytically inactive SARM1(E642A) together with MitoDsRed (MitoDR) in order to track mitochondria. After 7 days, axons were severed and mitochondrial movement was imaged 0, 2, and 4 hr after injury. In SARM1 KO neurons expressing GFP, mitochondrial movement was unchanged 4 hr after injury. In contrast, the number of motile mitochondria declines precipitously between 2 and 4 hr after injury in SARM1 KO axons re-expressing SARM1.

Catalytically inactive SARM1(E642A) is expressed at similar levels to wild-type (WT) SARM1 (*Figure 1—figure supplement 1*), but did not result in loss of mobile mitochondria after injury (*Figure 1A and B*). Hence, the loss of mitochondrial mobility in injured axons is not only SARM1-dependent, but also SARM1 NADase-dependent. Injury-induced loss of mitochondrial potential is also SARM1-dependent (*Geisler et al., 2019*), and so here we investigated whether or not this effect also requires a functional SARM1 NADase. The mitochondrial potential is the driving force for ATP production and can be measured with the fluorescent dye TMRM (tetramethylrhodamine methyl ester). As with mitochondrial dynamics, expression of WT SARM1, but not SARM1(E642A), leads to a dramatic loss of mitochondrial membrane potential after injury (*Figure 2C and D*). The finding that SARM1 NADase activity is required for loss of mitochondrial membrane potential suggests that the SARM1-induced decline in cytosolic $NAD^+$ levels either directly or indirectly influences bioenergetics inside the mitochondria.

In addition to mitochondrial dysfunction, calcium homeostasis is also disrupted in injured axons, with a large influx of calcium hours after injury (*Adalbert et al., 2012*; *George et al., 1995*; *Ma et al., 2013*; *Wang et al., 2000*; *Yang et al., 2013*). Here, we test if this calcium influx requires SARM1 enzymatic activity. We used Fluo-4, a calcium-sensitive fluorescent dye, to assess axonal calcium four hours after injury, the time point by which mitochondrial mobility and potential are disrupted. Axons show no calcium rise in SARM1 KO neurons expressing either GFP or SARM1(E642A), but have a significant increase in calcium when expressing WT SARM1 (*Figure 1E and F*). Hence, the loss of calcium homeostasis in injured axons also requires SARM1 NADase activity. Taken together, these findings support the view that SARM1 enzyme activity is essential for not only axon degeneration (*Essuman et al., 2017*), but also for the major proximate events that occur in injured mammalian axons.

## Manipulating either intracellular or extracellular calcium is ineffective in preserving injured axons

Having demonstrated that calcium influx into injured axons requires SARM1 NADase activity, we next explored the role of calcium influx in axonal demise. Prior studies argued that calcium release through the mitochondrial permeability transition pore (MPTP) (*Barrientos et al., 2011*; *Villegas et al., 2014*) or extracellular calcium influx (*Vargas et al., 2015*; *Wang et al., 2000*; *Witte et al., 2019*; *Yong et al., 2020*) are key drivers of axon degeneration. To test whether the MPTP has a role in axon degeneration, we incubated embryonic DRGs neurons with the MPTP inhibitor (Cyclosporin A, CsA), axotomized, and assessed the progression of axon degeneration and the rise in intracellular calcium. In contrast to prior findings, we observed no delay in the timing of axon degeneration with CsA treatment (*Figure 2A*). We also assayed the increase in intracellular calcium after injury and again found no significant effect of CsA (*Figure 2B*). To further explore the role of intracellular calcium, we incubated DRG neurons with 10 µM BAPTA to chelate intracellular calcium for 30 min prior to axotomy. This treatment had no influence on the progression of axon degeneration (*Figure 2C*), and was also unable to fully buffer the influx of extracellular calcium that occurs hours after injury (*Figure 2—figure supplement 1*). These results indicate that internal calcium is not a major determinant of injury-induced axon degeneration in this system; however, it may play a role in scenarios where SARM1 is less potently activated (*Li et al., 2021*).

While we found no clear role for intracellular calcium, there are numerous studies highlighting the importance of extracellular calcium for injury-induced axon degeneration (*Gerdts et al., 2011*; *Mishra et al., 2013*; *Ribas et al., 2017*; *Vargas et al., 2015*; *Wang et al., 2000*; *Yang et al., 2013*). As such, we explored the impact of chelating extracellular calcium on the progression of axon degeneration. In agreement with prior studies, we find that pre-incubation with 3 mM EGTA potently blocked injury-induced calcium influx and maintained morphologically intact axons for up to 48 hr after axotomy (*Figure 2D and E*; *Figure 2—figure supplement 1*). To explore when the influx of extracellular calcium is required, we treated with EGTA at the time of axotomy and then performed washout after 2 hr, or added the EGTA 2 hr after axotomy. The presence of EGTA from 0 to 2 hr after axotomy had no impact on the timing of axon fragmentation, while addition of EGTA 2 hr post-axotomy was as protective as treatment at the time of axotomy (*Figure 2D and E*). Hence, late influx of extracellular calcium is critical for axon degeneration in DRG axons, a finding consistent with previous reports (*Vargas et al., 2015*; *Witte et al., 2019*).

While these results are consistent with prior work highlighting the importance of extracellular calcium for axon degeneration, we did observe that severed axons developed prominent axonal

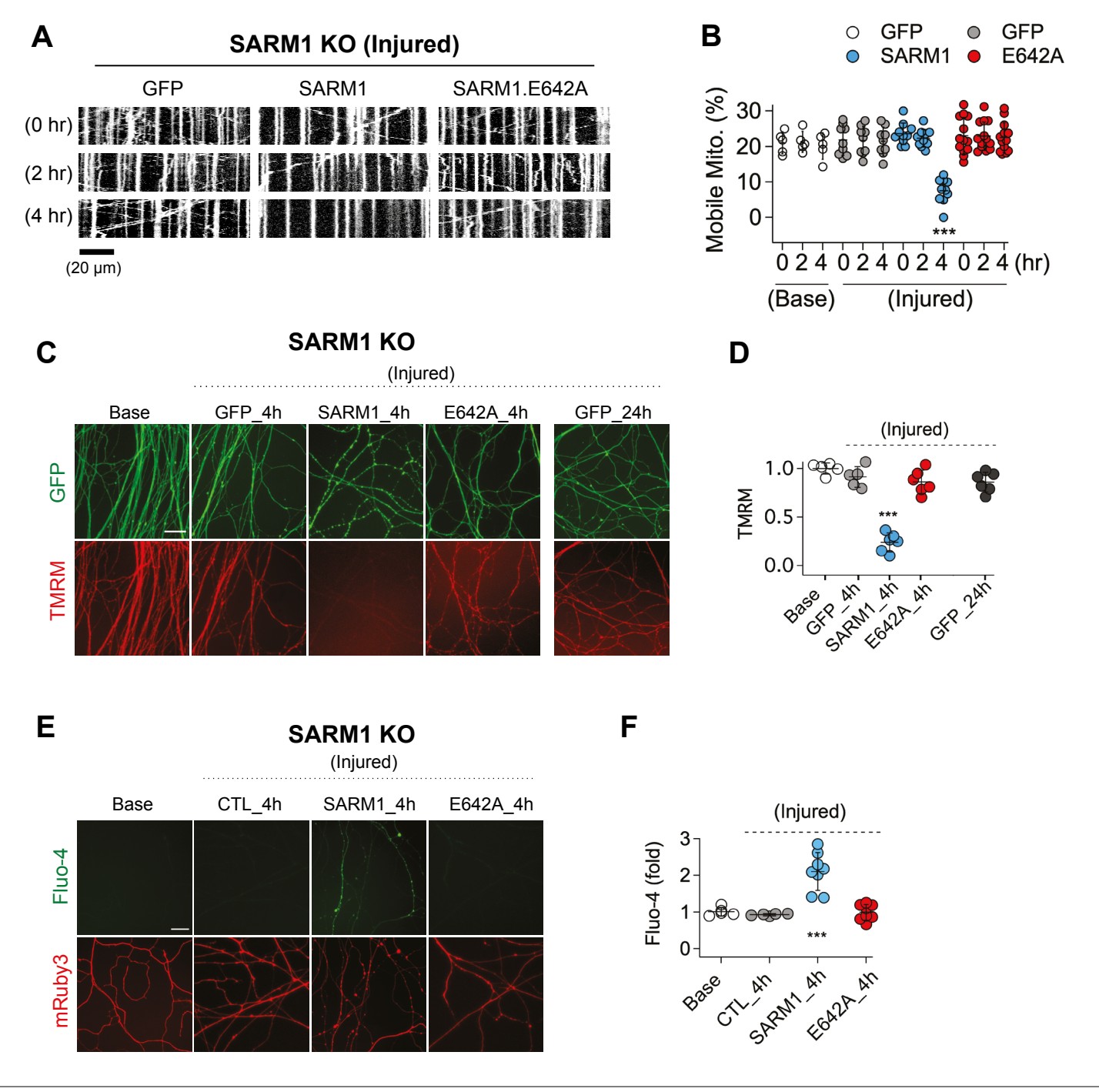

**Figure 1.** SARM1 enzymatic activity regulates mitochondrial movement and calcium homeostasis in injured axons. (**A**) Representative kymograph of injured SARM1 KO axons expressing either GFP, SARM1, or SARM1.E642A (E642A). For imaging mitochondria movement, MitoDsRed lentivirus was transduced in all experimental conditions. Live cell imaging was performed at different times (0, 2, or 4 hr) after axon injury. Scale bar = 20 μm (**B**) Quantification of mobile mitochondria for the neurons in (**A**). Data represent the mean ± SEM; n = 5 ~ 13 axons for each condition; one-way ANOVA with post hoc Tukey test, $F_{(11,99)} = 12.28$, $p < 0.0001$; NS, not significant; *, $p < 0.05$; **, $p < 0.01$ and ***, $p < 0.001$ (**C**) Representative images of mitochondrial potential imaged with 50 nM TMRM fluorescent dye in SARM1 KO axons expressing either of GFP, SARM1, or SARM1.E642A. Live-cell imaging was performed at the indicated times (0, 4, or 24 hr) after axon injury. Scale bar = 30 μm (**D**) Quantification of TMRM intensity from the experiment in (**C**). Injured SARM1 KO axons expressing the enzymatically disabled SARM1 mutant (E642A) maintained TMRM signal without significant loss. Data represent the mean ± SEM; n = 5 ~ 6 embryos for each condition; one-way ANOVA with post hoc Tukey test, $F_{(4,23)} = 53.11$, $p < 0.0001$; NS, not significant; *, $p < 0.05$; **, $p < 0.01$ and ***, $p < 0.001$ (**E**) Representative images of calcium influx imaged with 1 μM Fluo-4 fluorescent dye in SARM1

*Figure 1 continued on next page*

*Figure 1 continued*

KO axons expressing either of GFP, SARM1, or SARM1.E642A. Live-cell imaging was performed at different times (0, or 4 hr) after axon injury. Scale bar = 30 μm (**F**) Quantification of Fluo-4 intensity from the experiment in (**E**). Injured SARM1 KO axons expressing the enzymatically disabled SARM1 mutant (E642A) completely prevent calcium influx. Data represent the mean ± SEM; n = 5 ~ 8 embryos for each condition; one-way ANOVA with post hoc Tukey test, $F_{(3,22)}$ = 23.05, $p < 0.0001$; NS, not significant; *, $p < 0.05$; **, $p < 0.01$ and ***, $p < 0.001$.

The online version of this article includes the following figure supplement(s) for figure 1:

**Figure supplement 1.** Expression level of SARM1.WT and SARM1.E642A.

swellings following extracellular calcium chelation (*Figure 2D*, red arrowheads). Moreover, previous work showed that SARM1 is activated within two hours after injury and leads to NAD+ depletion and metabolic catastrophe (*Sasaki et al., 2020a*), raising the question of whether blocking the later influx of calcium maintains axons in a healthy state. To investigate this question, we assessed mitochondrial membrane potential and mobility in injured axons treated with EGTA. We find that mitochondria potential is lost by four hours post-axotomy whether or not EGTA is present (*Figure 2F and G*). Similarly, EGTA treatment fails to maintain mitochondrial mobility after axotomy (*Figure 2H and I*). Indeed, treatment with EGTA halts mitochondrial movement and disrupts mitochondrial depolarization (*Figure 2—figure supplement 1*) after 4 hr even in the absence of injury, demonstrating that chelating extracellular calcium is not an effective method to maintain healthy axons, and instead disrupts normal axonal physiology. These findings are in contrast to loss of SARM1, which maintains both mitochondrial potential and mobility after axotomy (*Figure 1*). Taken together, these findings suggest that prior studies showing protection of axons by blocking the large calcium influx that occurs prior to degeneration were likely maintaining axonal structure but not axonal physiology, and suggest that the key role for calcium influx may be to trigger fragmentation of metabolically non-functional axons.

## Live single axon imaging defines the temporal relationship between calcium influx and axonal fragmentation

To quantitatively assess the relationship between this late influx of calcium and axonal fragmentation, we developed a live imaging system with single axon resolution (*Figure 3—figure supplement 1*). This allows us to assess temporal relationships of the asynchronous axon degeneration process that is not possible in mass cultures. We used lentivirus to co-express GCaMP6 and mRuby3 in cultured DRG neurons to monitor calcium influx and axon morphology simultaneously. After 7 days in culture, when both proteins were strongly expressed, we performed axotomy by focusing laser light on a 1 × 1 μm region containing a single axon (*Figure 3A*). Images of the distal axon were repeatedly acquired until the injured axon degenerated. *Figure 3B* and *Video 1* show the progression of calcium influx and axon degeneration for a single axon, features that are representative of all the injured axons analyzed. Immediately after axotomy, a first peak of calcium bidirectionally propagates along the axon from the injury site (white triangle). This elevated intracellular calcium is rapidly cleared, demonstrating that calcium homeostasis functions normally at this time, and is consistent with prior work and the findings in *Figure 2E* that the initial influx of calcium does not contribute to axon fragmentation (*Adalbert et al., 2012*; *Loreto et al., 2015*; *Vargas et al., 2015*). After nearly 4 hours, there is a large second influx of calcium that precedes any obvious change in axonal morphology. Soon thereafter, the axon thins and small swellings appear, then the axon swellings enlarge, and finally the axon fragments (*Figure 3B and C*). From analysis of 22 single axons, we found that the time to the appearance of the second calcium peak varied dramatically, from less than four hours to nearly 10 hr after axotomy (*Figure 3D*). However, once the second peak of calcium appeared, the axon fragmented soon thereafter. The correlation between the initiation of the second peak of calcium and axon degeneration was very strong (*Figure 3D*), with degeneration occurring ~100 min after initiation of calcium influx. In contrast, neither the duration nor intensity of the second calcium peak was well correlated with the timing of axon degeneration (*Figure 3E and F*). The very tight correlation between the initiation of the second influx of calcium and axonal fragmentation is consistent with the hypothesis that this calcium triggers the final disintegration of the axon.

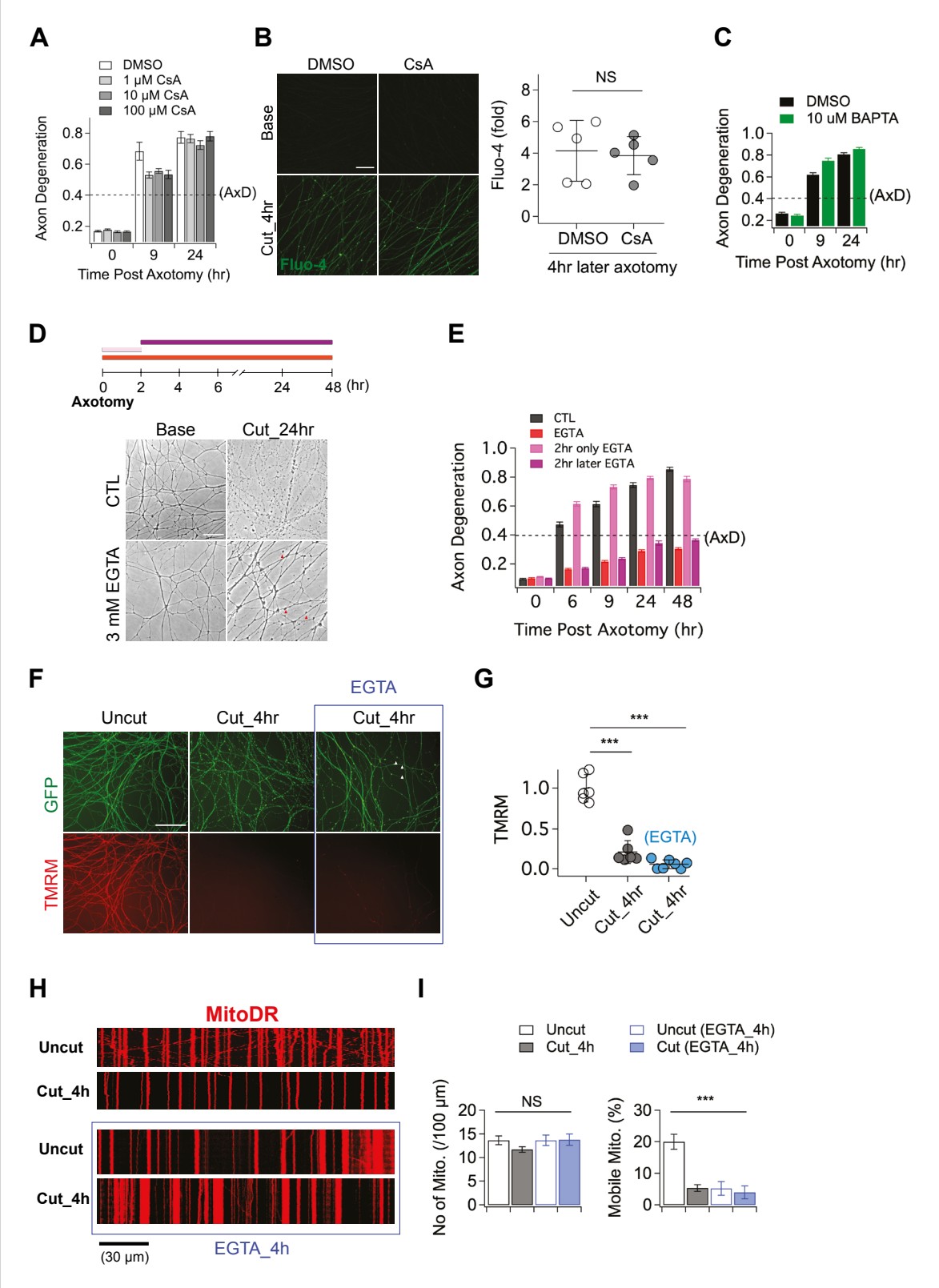

**Figure 2.** The role of calcium in axon degeneration. (**A**) Pre-incubation of MPTP inhibitor (1, 10, or 100 µM CsA) did not significantly prevent the degeneration of wild-type axons after axon injury. Axon degeneration is defined as a degeneration index >0.4 (dashed line). n = 3 embryos for each condition. (**B**) (Left) Representative images of calcium influx acquired with 1 µM Fluo-4 dye. Scale bar = 30 µm (Right) The degree of calcium influx in CsA pre-incubated injured axons is not significantly different from DMSO pre-incubated injured axons. Fold increment of injured axons at 4 hr after axotomy

*Figure 2 continued on next page*

*Figure 2 continued*

is calculated from uninjured axons. Data represent the mean ± SEM; n = 5 embryos for each condition; two-tailed unpaired t test, p = 0.39; NS, not significant; *, p < 0.05; **, p < 0.01 and ***, p < 0.001 (**C**) Internal calcium chelator (pre-incubation with 10 µM BAPTA) did not delay axon degeneration after axonal injury. n = 3 embryos for each condition. Data represent the mean ± SEM; Axon degeneration is defined as a degeneration index >0.4 (dashed line). (**D**) (Top) Experimental design. Extracellular calcium chelator, 3 mM ETGA, was included in the culture medium at different time points (2 ~ 48 hr vs 0 ~ 2 hr vs 0 ~ 48 hr). For addition of EGTA from 0 ~ 2 hr, culture medium was replaced at 2 hr. (Bottom, left) Representative bright-field images of axons. (**E**) Quantification of axon degeneration for the experiment in (**D**). Although there are axonal swellings (red triangle) in the presence of EGTA, injured axons remain intact when the EGTA is present 2 hr after axotomy. n = 3 embryos for each condition. Data represent the mean ± SEM; Axon degeneration is defined as a degeneration index >0.4 (dashed line). (**F**) Representative images of mitochondria potential (TMRM fluorescent dye) and axon morphology (GFP lentivirus) in uncut axons and cut axons ±EGTA. Scale bar = 100 µm (**G**) Quantification of the TMRM staining for the experiment in (**F**). EGTA incubation in injured axons does not maintain mitochondrial hyperpolarization. Data represent the mean ± SEM; n = 6 ~ 7 embryos for each condition; one-way ANOVA with post hoc Tukey test, F(2,16) = 98.27, p < 0.0001; NS, not significant; *, p < 0.05; **, p < 0.01 and ***, p < 0.001 (**H**) Representative kymograph of uncut and cut axons ±EGTA as indicated. (**I**) Quantification of total number of mitochondria (left) and mobile mitochondria (right) for the experiment in (**H**). Data represent the mean ± SEM; n = 8 ~ 12 axons for each condition; one-way ANOVA with post hoc Tukey test, for mobile mitochondria F(3,24) = 16.34, p < 0.0001; for number of mitochondria, F(3,34) = 0.8787, p = 0.46; NS, not significant; *, p < 0.05; **, p < 0.01 and ***, p < 0.001.

The online version of this article includes the following figure supplement(s) for figure 2:

**Figure supplement 1.** EGTA efficiently blocks calcium influx.

## Mitochondrial dysfunction precedes calcium influx in injured axons

With a method established for live imaging of single axons, we next explored the temporal relationship between calcium influx and mitochondrial stalling and loss of potential. Calcium influx is a potent mechanism for stopping mitochondria (*Wang and Schwarz, 2009*), and so we predicted that calcium influx would occur before mitochondrial stalling. We used lentivirus to express MitoDR and GCaMP6 in DRG neurons and performed laser axotomy. We imaged mitochondrial movement with MitoDR every 5 s for 300 s followed by calcium measurements. Images from MitoDR and GCaMP6 were acquired until mitochondria stopped, at which point only GCaMP6 imaging continued until its level increased more than twofold (*Figure 4A*). As shown in *Figure 4A* and in contrast to expectations, mitochondrial movement stops before the influx of calcium. Analysis of this single axon shows that fewer mitochondria are moving 3 hr after axotomy (*Figure 4B*), and that all mitochondrial are stalled by 4.4 hr after injury (*Figure 4B*, inset). At this point, the GCaMP6 signal is unchanged from baseline, but begins to rise soon thereafter. Quantitative analysis of single injured axons demonstrates that loss of mitochondrial mobility precedes calcium influx in each case, and this occurs ~25 min after mitochondria stop (*Figure 4B and C*). Hence, loss of mitochondrial mobility in injured axons cannot be due to calcium influx.

Next, we addressed the relationship between loss of mitochondrial potential and calcium influx. Calcium overload can induce loss of mitochondrial potential (*Abramov et al., 2007*)—if this occurs in injured axons, then we expect calcium influx to precede mitochondrial depolarization. We expressed GCaMP6 in DRG neurons, incubated the neurons with TMRM to assess mitochondrial potential, and imaged single axons in both channels every ten minutes until axons degenerated (*Figure 4D*). This imaging frequency was chosen to avoid photobleaching. The images and analysis from a single axon (*Figure 4D and E*) demonstrate that the fluorescent intensity of TMRM is fairly stable at baseline, then has an initial drastic drop (dark red dot in *Figure 4E*) followed by a steady decline (*Figure 4E and F*, red dots). In contrast, calcium has an abrupt rise (dark green dot), quickly reaching a higher steady-state level (*Figure 4E and F*, green dots). In the example shown, the abrupt drop in mitochondrial potential occurs one frame prior to the large increase in calcium. To assess this across axons, we identified the time at which the TMRM and GCaMP6 signal showed the largest frame-to-frame variation (*Figure 4F and G* and *Figure 4—figure supplement 1*). We found that the drastic drop of TMRM intensity occurs one frame prior to the large increase in calcium in five out of seven axons (71.43%), while these changes occurred during the same frame in two out of seven axons. The images are taken ten minutes apart, and so these data indicate that in injured axons mitochondria begin depolarizing prior to the calcium influx (*Figure 4G*). Moreover, these data, in conjunction with the analysis of mitochondrial mobility and calcium influx above, demonstrate a sequence of events in which first mitochondria stop, then begin losing their potential, and after that calcium enters the axon which subsequently degenerates.

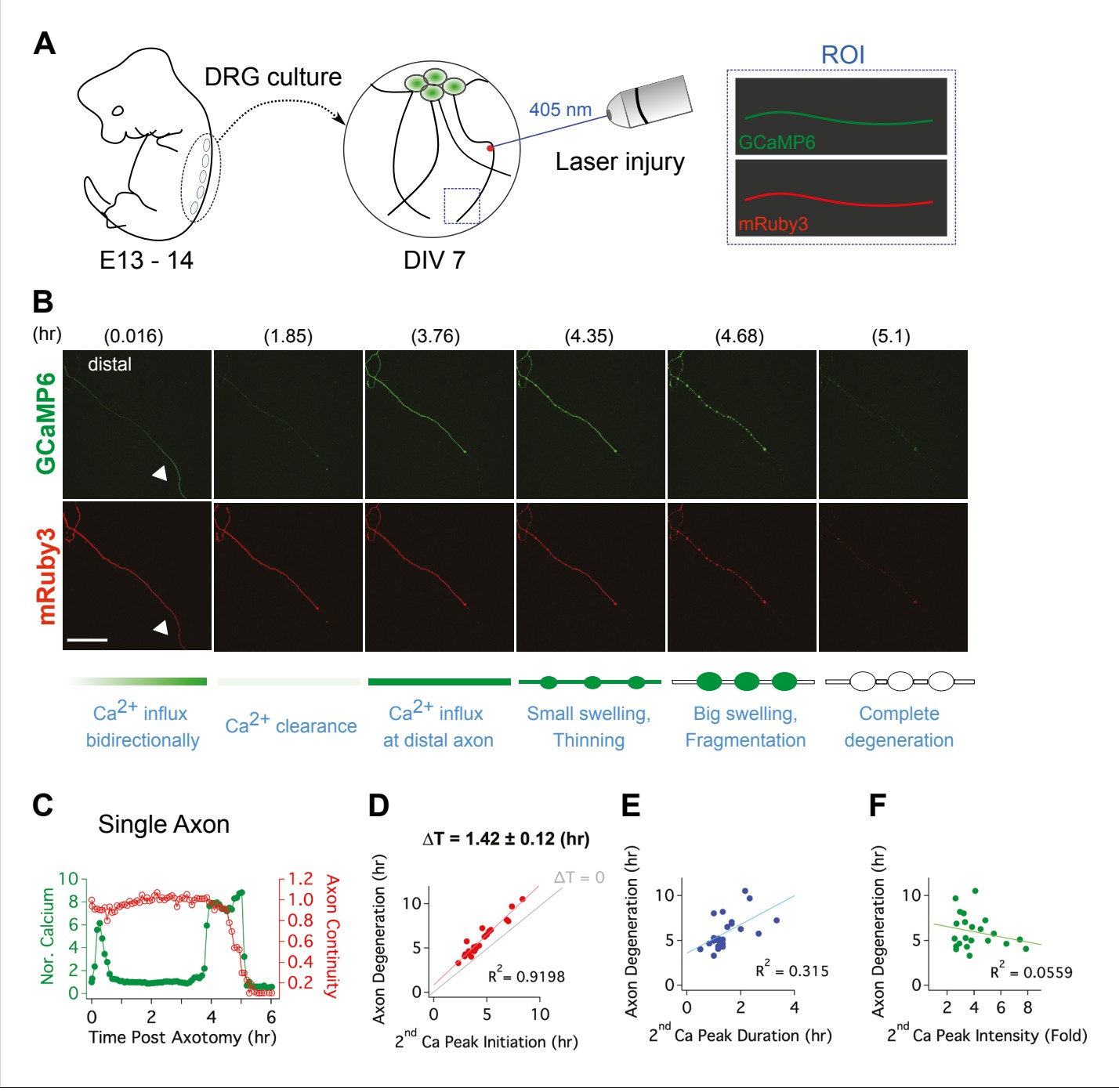

**Figure 3.** Live single axon imaging enables temporal dissection of cellular events in injured axons. (**A**) Schematic diagram of laser axotomy in cultured embryonic DRG neurons. GCaMP6 and mRuby3 were expressed to observe calcium fluctuations and axonal morphology. (**B**) Snapshots of an injured wild-type axon. Also see *Video 1*. Progression of axon degeneration is described at the bottom of the schematic. Note that there is both an early and late phase of calcium influx. The first peak of calcium occurs at the injury site (white triangle) before calcium levels return to normal. The second calcium peak persists until the axon degenerates. Scale bar = 100 µm (**C**) Representative analysis of a single injured axon. The calcium response (left y-axis) and measure of axon continuity (right y-axis) for a single axon is plotted over time after axonal injury. Note the two distinct calcium peaks.

The online version of this article includes the following figure supplement(s) for figure 3:

**Figure supplement 1.** DRG neuron culture for single axon imaging.

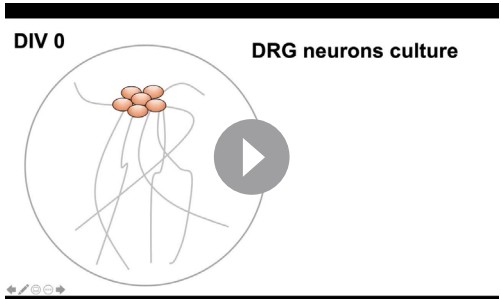

**Video 1.** Time lapse imaging of injured single axon.
https://elifesciences.org/articles/71148/figures#video1

## ATP loss precedes mitochondrial stalling

Calcium influx cannot explain the loss of mitochondrial mobility, so we considered other potential mechanisms. Once the SARM1 NADase is activated, ATP is lost soon thereafter (*Gerdts et al., 2015*) because NAD⁺ is required for ATP synthesis via both glycolysis and oxidative phosphorylation. The loss of ATP in injured distal axons is fully SARM1 dependent (*Figure 5—figure supplement 2*). Mitochondria are transported by ATP-dependent molecular motors, and so we hypothesized that loss of ATP may cause loss of mitochondrial mobility. To assess this hypothesis, we explored the relationship between ATP loss and mitochondrial stalling. We used lentivirus to express PercevalHR, a sensor of relative ATP levels (*Tantama et al., 2013*), and MitoDR in DRG neurons and performed laser axotomy. For the validation of PercevalHR in axons, we applied CCCP (Carbonyl Cyanide m-chlorophenyl hydrazone) to DRG neurons expressing PercevalHR. Using HPLC to measure axonal metabolites, we previously demonstrated that a 2 hr treatment with 50 µM CCCP lowers ATP by ~60 % in DRG axons (*Summers et al., 2014*). The same CCCP treatment leads to an ~70 % drop in the fluorescence level of PercevalHR (*Figure 5—figure supplement 1*), demonstrating that PercevalHR gives a reasonable estimate of changing ATP levels in axons. Upon injury, we found that ATP loss and loss of mitochondrial mobility temporally overlap, and so it was not possible to define a window after which one process was complete and the other had yet to start, as we did with calcium and mitochondrial dynamics above. Instead, we quantitatively assessed the relationship between the degree of ATP loss as defined by loss of the PercevalHR signal and the fraction of mitochondria that stop moving in injured axons. Based on ATP measurements from bulk injured axons, we knew that most ATP loss occurred between 3 and 4 hr after injury (*Figure 5—figure supplement 2*). Therefore, we imaged baseline mitochondrial mobility and relative ATP levels before injury, and then re-imaged the PercevalHR every 5 min for 3.5 hr after single axon injury. We then calculated the percent change in PercevalHR intensity from baseline. Immediately after the final imaging of PercevalHR at 3.5 hr, we imaged mitochondrial mobility by acquiring images every 5 s for 300 s and calculated the percent drop in the fraction of motile mitochondria compared to baseline (*Figure 5A and B*). In every axon, the percent drop in the ATP sensor was larger than the percent drop in the fraction of mobile mitochondria (*Figure 5B* left), and there was a strong correlation between the extent of ATP loss and mitochondrial stalling (*Figure 5B* right; $R^2 = 0.61$, n = 9). We continued to measure mitochondrial mobility from 3.5 hr after injury until mitochondrial movement ended. We found a strong inverse correlation between the extent of ATP loss at 3.5 hr and the remaining time until the complete loss of mitochondrial mobility (*Figure 5C*; $R^2 = 0.76$, n = 9). In other words, the extent of ATP loss by 3.5 hr after injury is a strong predictor of when mitochondrial will ultimately stop moving in an injured axon. All these results are consistent with the model that ATP loss causes loss of mitochondrial mobility. We did not repeat these results in SARM1 KO axons because we have already demonstrated that both ATP loss and loss of mitochondrial mobility are fully SARM1 dependent (*Figure 5—figure supplement 2* and *Figure 1*).

## Calcium is required for loss of membrane integrity during axon fragmentation

Having shown that calcium influx is a late event in the axon degeneration process, we assessed the temporal relationship between calcium influx and two other late events, loss of membrane lipid asymmetry and loss of membrane integrity (*Yong et al., 2020*). In healthy membranes, phosphatidylserine is preferentially found in the inner leaflet of the plasma membrane. In cells undergoing apoptosis and in degenerating axons, phosphatidylserine is exposed on the outer leaflet where it serves as an 'eat-me' signal to phagocytic cells (*Sapar et al., 2018*; *Segawa and Nagata, 2015*; *Shacham-Silverberg et al., 2018*; *Wakatsuki and Araki, 2017*). To assess the temporal relationship among calcium influx, loss of membrane asymmetry, and axon fragmentation, we expressed GCaMP6 and

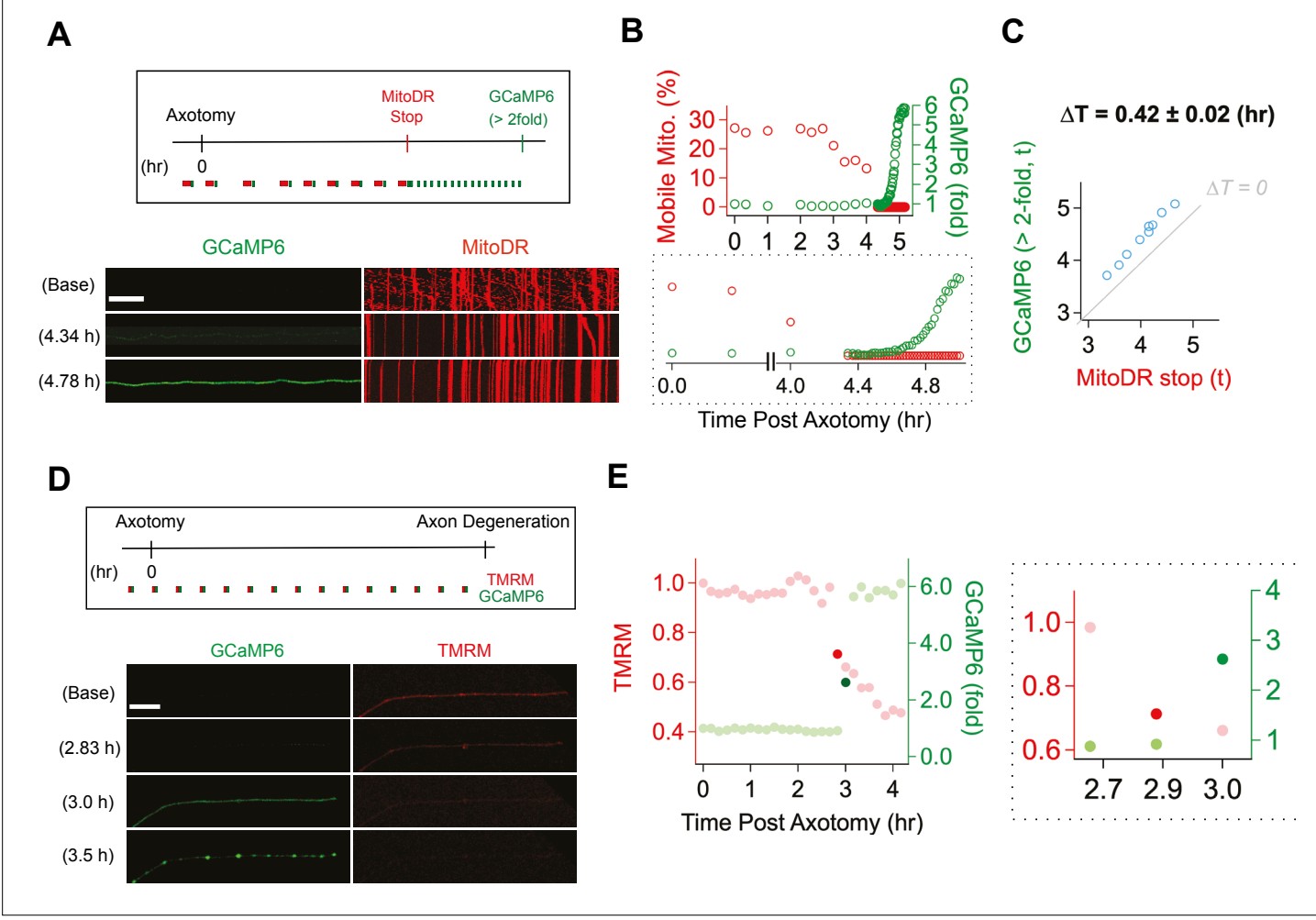

**Figure 4.** Mitochondrial dysfunction precedes calcium influx in injured axons. (**A**) (Top) Experimental design for observing calcium influx (GCaMP6) and mitochondria movement (MitoDR) after axon injury. MitoDR images were acquired every 5 s (for 300 s, 60 frames) followed by GCaMP6 imaging. Once the mitochondria in that axon stopped, GCaMP6 images were then acquired once/minute. (Bottom) Representative images of GCaMP6 and kymograph at the indicated times. Note that mitochondria stop prior to calcium influx. Scale bar = 30 μm (**B**) (Top) Single axon analysis after injury. The percentage of mobile mitochondria (red, left y-axis) and the fold change in calcium (green, right y-axis) for a single axon were plotted over time after axonal injury. (Bottom) Inset from graph highlights that mitochondria stop moving before calcium levels rise. (**C**) Group data from single axons show that the time difference (ΔT) between cessation of mitochondrial mobility and calcium influx, defined as a twofold increase from baseline, is ~0.42 ± 0.02 hr, indicating that mitochondria stop before calcium influx in injured axons. The gray line (ΔT = 0) shows expected results if mitochondria stopped and calcium influx occurred simultaneously. n = 9 axons. (**D**) (Top) Experimental design to observe calcium influx (GCaMP6) and loss of mitochondrial potential (TMRM) after axonal injury. GCaMP6 and TMRM were imaged every 10 min until axon fragmentation. (Bottom) Representative images shown at the indicated times. The TMRM signal declines by 2.83 hr after axonal injury, while calcium influx does not occur until 3.0 hr after injury. Scale bar = 30 μm (**E**) (Left) Analysis of the single axon in D. The ratio of TMRM signal from baseline (left y-axis) and the fold increment of calcium (right y-axis) were plotted over time after axonal injury. (Right) The enlarged insight highlights the point at which there is a dramatic change in the mitochondria potential (brighter red dot) and calcium levels brighter green dot. Note that the change in TMRM from baseline precedes the change in calcium. This was observed in 5 out of 7 axons, while in 2 out of 7 axons the change occurred in the same 10 min imaging bout.

The online version of this article includes the following figure supplement(s) for figure 4:

**Figure supplement 1.** Assessing mitochondrial potential and calcium influx in injured axons.

mRuby3 in DRG neurons, laser axotomized, and incubated with Alex Fluor 647-conjugated Annexin-V, which binds extracellular phosphatidylserine (*Sievers et al., 2003*). In the example shown, calcium rises first, followed by staining with Annexin-V, and soon thereafter the mRuby3 signal declines indicative of axon fragmentation (*Figure 6A*). Indeed, in all cells analyzed calcium influx preceded phosphatidylserine exposure, and occurred ~0.51 ± 0.04 (hr) prior to Annexin-V staining (*Figure 6B*). Calcium can inhibit the ATP-dependent flippase that maintains phosphatidylserine on the inner leaflet (*Bitbol*

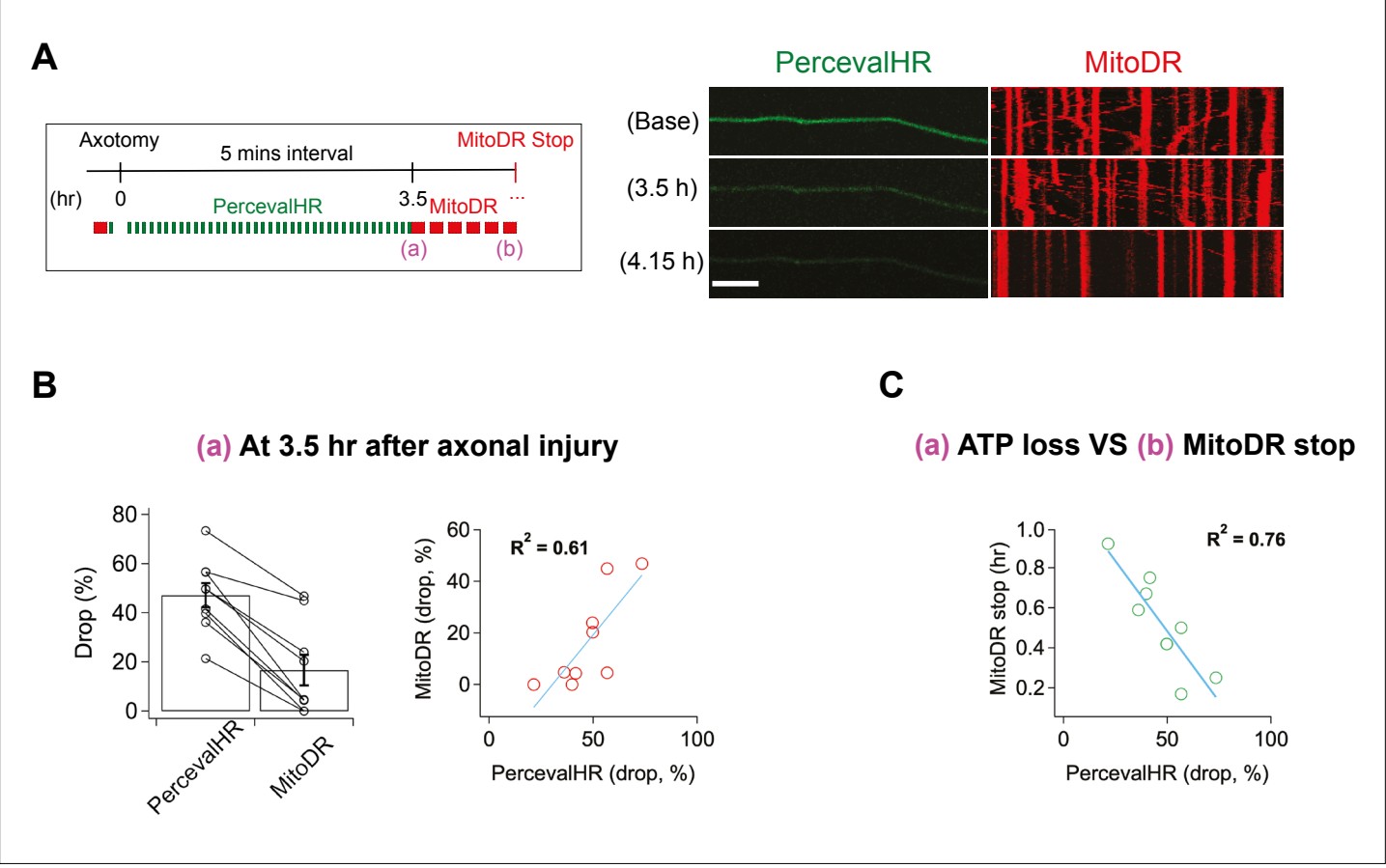

**Figure 5.** ATP levels drop before mitochondria stop in injured axons. (**A**) (Left) Experimental design for imaging changes to ATP (PercevalHR) and mitochondrial movement (MitoDR) after axonal injury. Prior to axotomy, baseline PercevalHR intensity and mitochondrial movement were measured. PercevalHR was imaged every 5 min until 3.5 hr after axonal injury, while mitochondria were imaged every 5 min starting 3.5 hr after axotomy until movement ceased. (Right) Representative images for PercevallHR and kymographs of moving mitochondria at the indicated times. Scale bar = 30 μm (**B**) (Left) Percentage decline from baseline at 3.5 hr post-axotomy for PercevalHR intensity and for the fraction of motile mitochondria. Lines connect data for individual cells. (Right) Linear regression plot of group data. n = 9 axons. (**C**) The percentage decline of PercevalHR intensity at 3.5 hr after axonal injury is plotted against the subsequent time until mitochondria stop moving for that axon. Linear regression plot of group data. n = 9 axons.

The online version of this article includes the following figure supplement(s) for figure 5:

**Figure supplement 1.** Validation of PercevalHR.

**Figure supplement 2.** Axonal ATP level in WT and SARM1 KO after axotomy.

---

*et al., 1987*; *Soupene, 2008*), and so the influx of calcium and/or the decline in ATP likely triggers the loss of membrane asymmetry during axon degeneration. We wished to test the role of calcium by blocking influx with EGTA; however, this experiment is not possible because Annexin-V binding to phosphatidylserine requires extracellular calcium.

Next, we assessed the relationship among SARM1, calcium influx, and the loss of membrane integrity in injured axons. To assess membrane integrity, we applied fluorescently labeled macromolecules (3 kDa dextran) to neurons expressing cytosolic GFP. In uninjured neurons, cytosolic GFP fills the axon while the dextran is excluded (*Figure 6C*). After injury, axon swelling is apparent, and mitochondrial localize to these swellings (*Figure 7—figure supplement 1*). The swellings retain GFP and still exclude dextran. Later, discrete puncta of dextran appear in axonal fragments, and such fragments contain no visible GFP. We interpret this as axon swellings that burst, releasing soluble GFP and allowing entry to the high-molecular-weight dextran (*Figure 6C*). Next, we compared dextran uptake in injured axons from wild type and SARM1 KO neurons, as well as wild-type neurons treated with EGTA. By 4 hr after axotomy of wild-type neurons, dextran is present throughout the axons, indicative of a loss of membrane integrity. In axotomized SARM1 KO neurons, dextran is excluded from axons

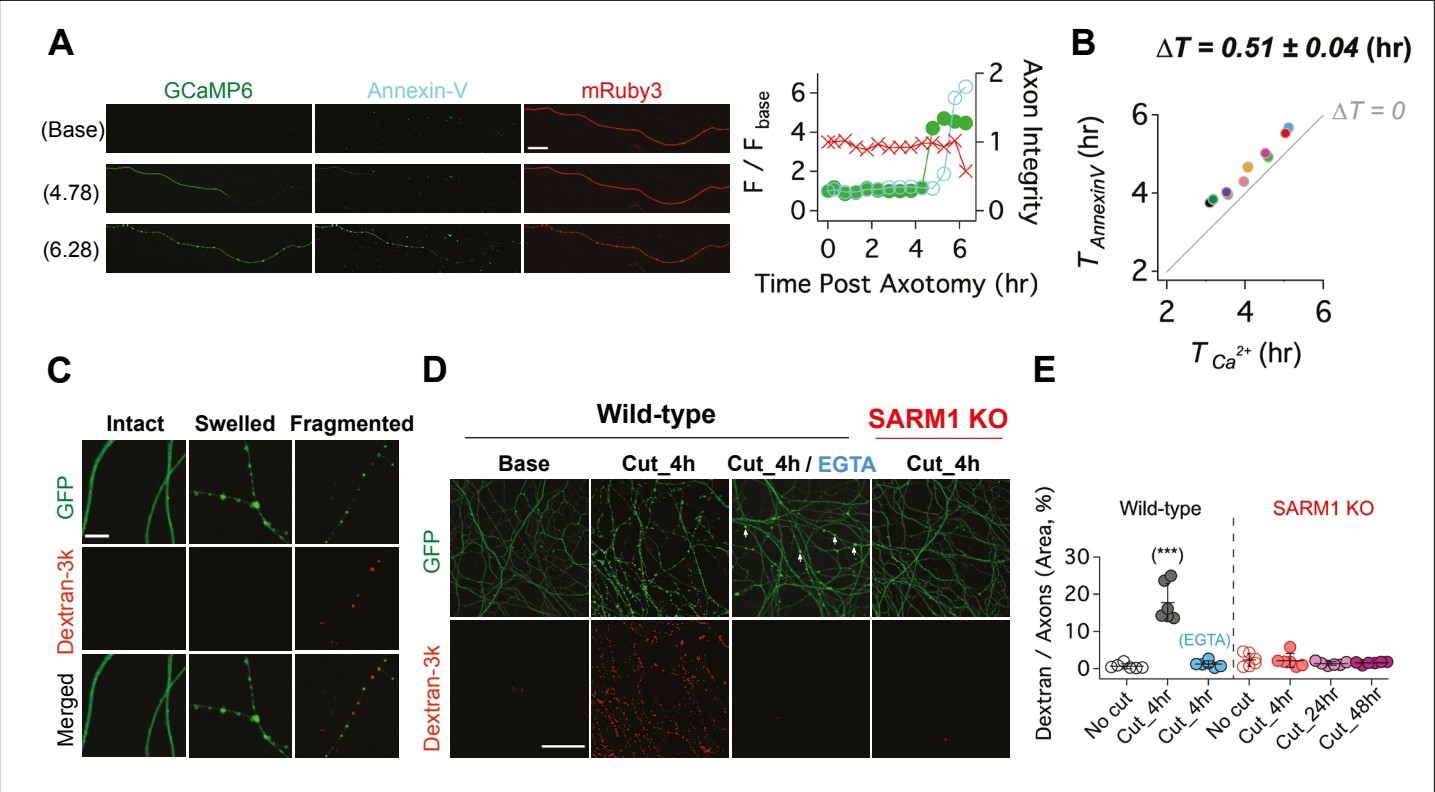

**Figure 6.** Calcium influx disrupts membrane integrity. (**A**) (Left) Snapshots of representative live axon images for GCaMP6, mRuby3, and Alex647-conjugated Annexin-V at baseline, and 4.78 and 6.28 hr after axotomy. (Right) Representative single axon analysis. Y-axis (left) is plotted by the fold increase of fluorescent intensity (F / F $_{base}$) of either GCaMP6 (green color dots) or Annexin-V (cyan color dots) from the baseline after axon injury. Axon integrity (y-axis, right) is calculated by the relative mRuby3 intensity from the baseline. Note that calcium influx precedes Annexin-V exposure in an injured axon. Scale bar = 50 μm (**B**) After axotomy, the time until calcium influx is plotted vs the time until the rise in Annexin-V. Dashed line (ΔT = 0) represents the values if phosphatidylserine exposure (Annexin-V staining) and calcium influx occurred simultaneously. Calcium influx precedes phosphatidylserine exposure by an average of 0.51 ± 0.04 hr. n = 10 axons. (**C**) (Top) Representative images of intact, swollen, and fragmented axons during the process of axon degeneration. The axonal morphology is labeled with GFP that was transduced through GFP-lentivirus. Texas Red conjugated Dextran-3kDa was pre-incubated 30 min prior to image acquisition. Note that Dextran-3kDa is only observed in the fragmented axons, not in swollen axons. Scale bar = 10 μm (**D**) Representative images of GFP-expressing axotomized wild-type and SARM1 KO axons. The membrane impermeable Dextran-3k enters injured wild-type axons, but after injury is excluded from both EGTA-treated wild-type axons and SARM1 KO axons (GFP labels axons). Scale bar = 100 μm (**E**) Group data. Quantification of dextran-3k staining intensity in the indicated genotypes and times. Data represent the mean ± SEM; n = 6 embryos for each condition; one-way ANOVA with post hoc Tukey test, $F_{(6,35)}$ = 46.16, p < 0.0001; NS, not significant; *, p < 0.05; **, p < 0.01 and ***, p < 0.001.

for at least 48 hr. Interestingly, when wild-type neurons are incubated with EGTA, injured axons still swell (arrowheads, *Figure 6D*), but dextran is excluded (*Figure 6D and E*). Therefore, we conclude that calcium is necessary for the loss of membrane integrity and the morphological transition from axonal swelling to fragmentation.

## Discussion

In injured axons, the molecular function of SARM1 is well understood, but the ensuing molecular and cellular changes leading to axonal demise are much more poorly defined. SARM1 NADase activity is critical for the ultimate demise of injured axons (*Essuman et al., 2017*), and here, we show that this enzymatic activity is also required for intermediate phenotypes such as disrupted mitochondrial and calcium homeostasis. To explore the events that occur after NAD⁺ cleavage, we used live imaging with single axon resolution to investigate dynamic changes to ATP, mitochondria, calcium, and membranes. The data describe an ordered series of events beginning with loss of ATP, and followed by mitochondrial dysfunction, calcium influx, exposure of phosphatidylserine and loss of membrane permeability ultimately resulting in catastrophic axon fragmentation.

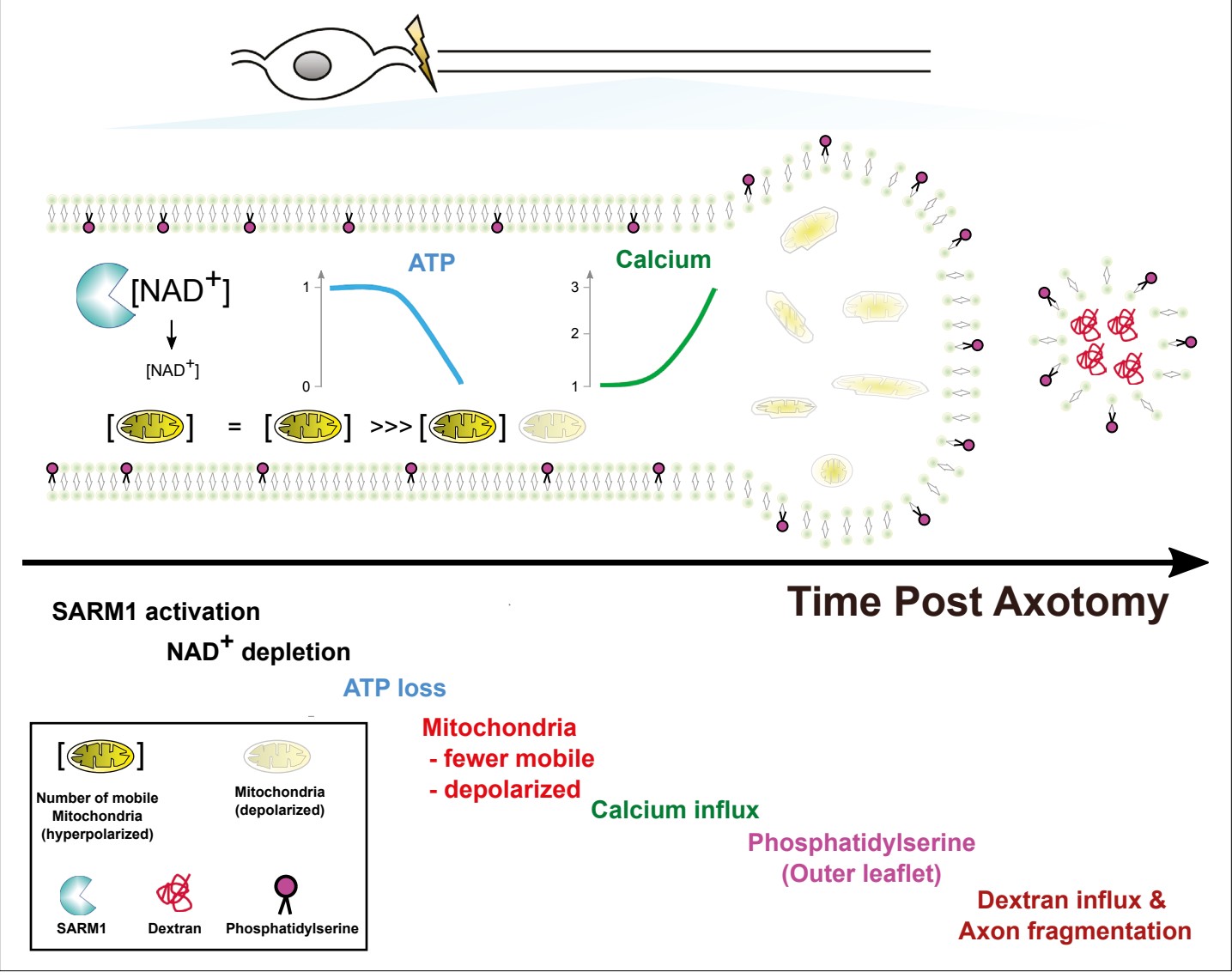

**Figure 7.** Model of axon degeneration.

The online version of this article includes the following figure supplement(s) for figure 7:

**Figure supplement 1.** Mitochondria accumulate in axonal swellings in injured axons.

In this study, we used live imaging of single axons to monitor structural and physiological changes during axon degeneration. While the absolute timing of cellular events varied dramatically from axon to axon, the relative timing was quite consistent. Hence, this method allowed us to order cellular events, which is not possible when averaging responses from many axons that are responding asynchronously. Our findings in conjunction with prior studies lead to a simple model of axon degeneration (*Figure 7*). Following activation of SARM1, NAD$^+$ is cleaved and its levels drop rapidly. Upon robust NAD$^+$ depletion, both glycolysis and oxidative phosphorylation will be impaired, and so ATP production will decline. This loss of ATP will impact molecular motors, leading to the observed halting of mitochondria. Soon after mitochondria halt, they lose their membrane potential. Since this occurs before calcium increases, this cannot be due to calcium overload. Instead, the loss of NAD$^+$ and ATP likely disrupts mitochondrial homeostasis. We next observed influx of extracellular calcium. The loss of ATP is a likely culprit, as ionic pumps require ATP and their loss will lead to disrupted calcium extrusion, membrane depolarization, and calcium influx. Many prior studies have attempted to define the source of calcium influx (*George et al., 1995*; *Gerdts et al., 2011*; *Stys, 2005*; *Villegas et al., 2014*; *Yang et al., 2013*), and have provided evidence for many candidates including voltage-gated ion

channels, ion pumps, and ion exchangers in both intracellular organelles and the plasma membrane. We speculate that many candidates participate because the loss of ATP disrupts many aspects of calcium homeostasis. Here, we have only focused on the large influx of calcium that occurs soon before degeneration. Our study does not speak to the potential role of smaller fluctuations in calcium that may occur earlier in the process. After this large influx of calcium, the subsequent exposure of phosphatidylserine to the outer leaflet of the membrane is likely due to the failure of lipid flippases to maintain asymmetry. Since these flippases are ATP-dependent and can be inhibited by calcium (*Bitbol et al., 1987*; *Pomorski and Menon, 2006*; *Soupene, 2008*), loss of ATP and calcium influx may both contribute to the externalization of phosphatidylserine. The ultimate loss of membrane integrity, however, requires influx of extracellular calcium, as EGTA blocks fragmentation of injured axons. Such axons have extensive swellings and completely dysfunctional mitochondria. While calcium influx has long been assumed to the be the essential final step in axon loss, our findings suggest instead that calcium influx is merely leading to the morphological destruction of axons that are already physiologically dead. Instead, our findings highlight ATP loss as the likely point of no return for an injured axon, disrupting mitochondrial, calcium, and membrane homeostasis and thereby triggering axonal demise.

## Materials and methods

### Animals

All procedures were performed in accordance with guidelines mandated in the National Institutes of Health Guide for the Care and Use of Laboratory Animals and approved by the Washington University School of Medicine in St. Louis Institutional Animal Care and Use committee. CD1 mice (gestation day 11.5) for sensory neuron cultures were purchased from Charles River Laboratories, and SARM1KO mice were developed using homologous recombination in embryonic stem cells to delete exons 1 and 2, were backcrossed into a C57 background, and were a gift from M. Colonna at Washington University in St. Louis (*Szretter et al., 2009*). All neurons in this study were derived from a mix of male and female embryos at embryonic day 13.5 or 14.5.

### Western blot analysis

Lysate buffers (60 mM Tri-HCl, pH 6.8; 50 % glycerol; 2 % SDS; 0.1 % bromophenol blue) contain protease cocktail (cOmplete, mini, EDTA-free protease inhibitor; 1183617001, Millipore Sigma) and phosphatase inhibitor cocktail (P0044, Millipore Sigma). The lysates were precleared of debris by centrifugation at 10,000 g in a refrigerated microcentrifuge for 10 min. Supernatants were mixed with 5 % 2-mercaptoethanol (Millipore Sigma) and then boiled for 10 min. Antibodies used: Rabbit anti-Tubulin III (1:4000, Millipore Sigma); HRP conjugated anti-rabbit antibody (1:10,000, #111-035-045, Jackson ImmunoResearch); Rabbit-anti-SARM1 (1:1000, #13022, Cell signaling) and (1:5000); mouse anti-GFP (1:1000, #2955 S, Cell signaling); HRP-conjugated anti-mouse antibody (1:5000, 115-035-003, Jackson ImmunoResearch).

### TMRM / Fluo-4

A total of 50 nM TMRM (T668, Thermo Fisher Scientific) and 1 µM Fluo-4 (F14201, Thermo Fisher Scientific) were pre-incubated for 30 min prior to image acquisition. When Fluo-4 was incubated more than 2 hr, we found that the intensity of Fluo-4 suddenly increased even in the absence of injury, and then axons degenerated. So, we only used Fluo-4 to check the current status of calcium and finished the imaging session within 1 hr.

### Lentivirus construction/production

FUGW-PercevalHR (Addgene #49083) GCaMP6 and mRuby3 (*Ko et al., 2020*), human SARM1.WT and human SARM1.E642A (*Essuman et al., 2017*) and MitoDsRed (*Summers et al., 2014*) were transfected into HEK 293 cells for lentivirus production. Briefly, the cells were seeded at 70~80 % confluency per 35 mm well the day before transfection. The constructs (1.2 µg) were cotransfected with vesicular stomatis virus G (600 ng) and pSPAX2 (800 ng) using FuGENE 6 (Promega). The lentiviral supernatants were collected 2 days after transfection, and then the cleared supernatant was concentrated with Lenti-X Concentrator (Clontech) to a final concentration of 1 ~ 10 x 10$^7$ particles /

ml. Lentivirus transduction efficiency was monitored with tagged fluorophore and western blot analysis and is routinely ~100 % in DRG neurons.

## DRG neurons culture/experimental timeline

All plates for DRG cultures are coated with 0.1 mg/ml poly-D-lysine (Millipore Sigma) followed by laminin (3 µg/ml; Invitrogen). CD1 mouse and SARM1 KO DRG neurons were dissected from a mix of male and female embryos at embryonic day 13.5 or 14.5. They were incubated with 0.05 % trypsin containing 0.05 % EDTA at 37 °C for 20 min and then washed three times with DRG growth medium (neurobasal media from Gibco) containing 2 % B27 (Invtrogen), 50 ng/ml nerve growth factor (Harlan Laboratories), 1 µM 5-fluoro-2'-deoxyuridine (Millipore Sigma), 1 µM uridine (Millipore Sigma), and penicillin/ streptomycin (Thermo Fisher Scientific). The cell density of these suspensions was adjusted to ~7 x 10$^6$ cells/ml.  Twoµl suspensions were placed in 24-well plates (Corning) for western blots and axon degeneration assays, Chamber slides (Nunc Lab-Tek, Thermo Fisher Scientific) were used for immunocytochemistry and FluoroDish (FD35-100, World Precision Instruments) were used for live single axon imaging. Lentivirus was transduced at one or 2 days in vitro (DIV). At DIV 7, assays for axon degeneration and/ or live axon imaging were performed.

## DRG neuron culture for single axon imaging

For single axon imaging, conventional 2 µl suspensions (~7 x 10$^6$ DRG cells / ml) lead to extensive overlap of DRG axons making it difficult to distinguish individual axons. Moreover, DRG neurons did not survive well in low-density culture (~7 x 10$^4$ cell / ml). To circumvent these problems, we plated two different densities of DRG neurons in one Fluorodish (*Figure 3—figure supplement 1*). Briefly, 2 µl suspensions (~7 x 10$^6$ cells/ml) were plated on one side of FluoroDish, and then 2 µl suspensions (~7 x 10$^4$ cells/ml) were thinly spread with a pipette tip on the other side of FluoroDish. This method provides the robust health of a high-density culture with the capacity to identify and image single axons.

## Live single axon imaging

DRG neurons were cultured in a glass bottom FluoroDish, enabling use of an immersion oil objective for calcium (GCaMP6), mitochondrial movement (MitoDR) and potential (TMRM), ATP (PercevalHR), and axon morphology (GFP or mRuby3). At DIV 2, lentivirus was transduced to cultured DRG neurons. Chemical dyes such as Annexin-V (#A23204, Thermo Fisher Scientific) and 3 kDa Dextran-Texas Red (#D3328, Thermo Fisher Scientific) were applied according to product instructions. Chamlide TC (Live Cell Instrument, South Korea) was used to maintain 37 °C temperature, 100 ml/min 5 % CO$_2$ /95 % airflow rate. A Leica DMI4000B microscope under confocal setting using 20 x oil immersion objective (NA 0.6) and Leica DFC7000 T 2.8 MP color microscope camera at RT was used under the control of the Leica Application Suite X software platform to acquire and analyze images. Optical sectioning and laser settings were kept constant across all image data acquisition sessions.

## Laser axotomy

Using a standard confocal microscope equipped with a 405 nm laser, a UV ablation method was utilized to selectively induce axonal injury of cultured DRG neurons in real-time (*Figure 3A*). To effectively induce laser axotomy of culture DRG neurons, a glass bottom ( < 0.17 mm) culture dish such as FluoroDish is necessary. 405 nm laser with 100 % intensity was used to induce laser axotomy with the FRAP (Fluorescence recovery after photobleaching) wizard in Leica application Suite X software. The injury site should be carefully chosen around the middle between the soma and axon terminal. If the injury site is close to soma, cell body death was often observed. If the injury site is too close to the distal axon then the immediate retraction of the injured axon results in too little residual axon for imaging.

## Mitochondria movement / kymograph analysis

For consecutive real-time imaging capture of mitochondria, images of MitoDR were recorded at 5 s intervals for a total of 60 frames by 558 (ex) / 583 (em)-nm laser at the designated time before and after laser axotomy. The mitochondria are considered mobile if the net displacement is more than 5 µm. Otherwise, they are defined as stationary.

## Axonal ATP measurement

For axonal metabolite measurement, axons and cell bodies were separated with microsurgical blade at 0–6 hours before metabolite collection. Culture plates were placed on ice and the culture medium was replaced with ice cold saline and then cell bodies were removed using a pipette. Saline was removed and replaced with 160 μL ice cold 50 % MeOH in water. After 5 min incubation, the solutions were transferred to microcentrifuge tubes containing 50 μl chloroform, shaken vigorously, and centrifuged at 20,000 x g for 15 min at 4 °C. The clear aqueous phase (140 μl) was transferred into a microfuge tube and lyophilized under vacuum. Lyophilized samples were reconstituted with 5 mM ammonium formate (15 μl), centrifuged (20,000 x g, 10 min, 4 °C), and 10 μl clear supernatants were analyzed with LC-MS/MS. Samples were injected into a C18 reverse phase column (Atlantis T3, 2.1 × 150 mm, 3 μm; Waters) using HPLC (Agilent 1290 LC) at a flow rate of 0.15 ml/min with 5 mM ammonium formate as mobile phase A and 100 % methanol as mobile phase B. Metabolites were eluted with gradients of 0–10 min, 0–70% B; 10–15 min, 70 % B; 16–20 min, 0 % B. The metabolites were detected with a triple quad mass spectrometer (Agilent 6,470 MassHunter; Agilent) under positive ESI multiple reaction monitoring using parameters for ATP (508 > 136, fragmentation (F) = 130 V, collision (C) = 30 eV, and cell acceleration (CA) = 4 V). Serial dilutions of standards for ATP in 5 mM ammonium formate were used for calibration. Metabolites were quantified by MassHunter quantitative analysis tool (Agilent) with standard curves and normalized by the axonal protein.

## Data acquisition and analysis

### Sample size (n)

In the figure legend, it is noted that sample size (n) means the number of axons or embryos. In single axon studies, only one axon of individual DRG neuron was imaged and analyzed (other branches of the same DRG neurons were not used). Otherwise, axons of cultured DRG neurons from individual embryo were imaged at least three times, and then averaged for the analysis.

### Axon degeneration

Axon degeneration is quantified based on axon morphology as the axon degeneration index (DI) using an ImageJ-based javascript (*Sasaki et al., 2009*). Axons should have less than 0.2 DI at baseline, otherwise, they were not used for the experiment. We define axon degeneration as an axon DI > 0.4.

### Calcium influx

The relative intensity of GCaMP6 from baseline was calculated as a measure of calcium influx. Given the interval of 5 ~ 10 min between images, the intensity change of GCaMP6 is variable. We routinely observed that the fluorescent intensity of GCaMP6 fluctuated ~10 % between image frames (*Figure 4—figure supplement 1*). While it is unclear whether or not these small fluctuations are functional, we did not see any corresponding changes in mitochondrial function, ATP or axon integrity. This study focuses on the major changes in calcium influx that occur prior to degeneration, and so we defined a twofold increase or greater of GCaMP6 intensity from the baseline as calcium influx.

### Axon continuity

The intensity of mRuby3 was used to monitor the intactness of axons. Because mRuby3 is a cytosolic protein, as the integrity and thickness of axonal membrane is narrowed and lost, the intensity of mRuby3 decreases. So, we defined more than 50 % intensity reduction of mRuby3 signal as the beginning of axon degeneration. When axonal fragmentation is observed, it is defined as a degenerated axon regardless of mRuby3 intensity.

### TMRM

After single axon injury, the intensity of TMRM were measured every 10 min (*Figure 4*). We found that there was a less 10 % fluctuation of fluorescent intensity between image frames. We calculated the percentage change of fluorescent intensity from the previous image (Diff_TMRM in *Figure 4—figure supplement 1*), and then defined a more than 30 % reduction as a significant loss of mitochondrial membrane potential.

## ATP sensor

PercevalHR was used in this study for measuring ATP fluctuation after axotomy. First, it was validated in *Figure 5—figure supplement 1*. Briefly, baseline PercevalHR signal in single axons was imaged. After laser axotomy, the fluorescent images were taken every 5 min for 3.5 hr. Later, each image frame was normalized to the baseline level for data analysis.

D-F. Grouped analysis from single axons for the (D) initiation, (E) duration, and (F) intensity of the 2nd peak of calcium compared to the time at which each axon fragments. The initiation of the second calcium peak occurs ~1.4 hr before and is strongly correlated with axon fragmentation, while the duration of the second peak is weakly correlated and the intensity of the second peak is not correlated with axon fragmentation. n = 22 axons.

The model depicts the ordered series of events that occur in an injured axon following SARM1 activation. These begin with NAD$^+$ loss, followed by ATP decline, loss of mitochondrial mobility, loss of mitochondrial polarization, influx of calcium, externalization of phosphatidylserine to the outer leaflet of the plasma membrane, and finally fragmentation of the axon allowing for influx of large molecular weight dextrans. Mitochondria localize to axonal swellings (see *Figure 7—figure supplement 1*).

Western blot analysis demonstrates that the lentiviral mediated expression of SARM1.WT and SARM1.E642A is very similar.

A)Treatment with 3 mM EGTA for 4 hr decreased mitochondrial membrane potential. TMRM staining in untreated and treated neurons after 4 hr. B). Statistical analysis of TMRM intensity using a t-test revealed a decrease in TMRM intensity (p = 0.0242, t = 2.579 df = 12, n = 7 embryos). C) Application of 3 mM EGTA and 10 µM BAPTA influence Calcium influx after axotomy. D) Pretreatment with 3 mM EGTA or 10 µM BAPTA decreases Calcium influx. Statistical analysis using a one-way ANOVA (F (2,15) = 19.04, p < 0.0001, n = 6 embryos) and Sidak's multiple comparison testing finds a decrease in calcium influx upon 3 mM EGTA (t = 6.109, p < 0.0001) or 10 µM BAPTA (t = 3.806, p = 0.0034) treatment.

For single axon imaging, 2 µl high-density cell suspensions (~7 x 10$^6$ cells / ml) were plated on the one side of a FluoroDish and 2 µl low-density cell suspensions (~7 x 10$^4$ cells / ml) were thinly spread on the opposite side. The high-density culture was required for maintenance of neurons in the low-density culture, and the low-density culture enabled imaging of single axons.

Axonal ATP was measured via LC-MS/MS at 0, 2, 4, and 6 hr post axotomy of cultured DRG neurons. Relative axonal ATP levels against 0 hr post axotomy were plotted for each genotype. Statistical analysis was performed by two-way ANOVA with Tukey multiple comparison (n = 12 derived from three independent cultures). F(1,88) = 79, p = 7.2 × 10$^{-14}$ between wild type (wt) and SARM1 knockout (SARM1 KO). *p < 1 × 10$^{-5}$ denotes a significant difference compared with WT at indicated time after axotomy.

Total 45 s video. (0 ~ 20 s) Brief demonstration of experimental design for single axon imaging. (21 ~ 45 s) Example of *Figure 3*. Briefly, image the baseline activity at distal axon, followed by axon injury with a 405 nm laser (blue square). Note that there is an increase of GCaMP6 intensity at the injury site, which is the first peak of calcium. Massive calcium influx enters the injured distal axon and then later axon starts degenerate. Image acquisition of both GCaMP6 and mRuby3 continues until the axon degenerates.

## Acknowledgements

We thank members of the DiAntonio and Milbrandt labs for fruitful discussions. This work was supported by National Institutes of Health grants R01CA219866 and RO1NS087632 (J M and A D), RF1-AG013730 (J M ), and F32NS117784 (L D)

## Additional information

### Funding

| Funder | Grant reference number | Author |
|---|---|---|
| National Institutes of Health | R01CA219866 | Jeffrey Milbrandt<br>Aaron DiAntonio |
| National Institutes of Health | RO1NS087632 | Jeffrey Milbrandt<br>Aaron DiAntonio |
| National Institutes of Health | RF1-AG013730 | Jeffrey Milbrandt |
| National Institutes of Health | F32NS117784 | Laura Devault |

The funders had no role in study design, data collection and interpretation, or the decision to submit the work for publication.

### Author contributions

Kwang Woo Ko, Conceptualization, Data curation, Formal analysis, Methodology, Validation, Visualization, Writing - original draft, Writing - review and editing; Laura Devault, Yo Sasaki, Data curation, Formal analysis, Visualization; Jeffrey Milbrandt, Aaron DiAntonio, Conceptualization, Funding acquisition, Resources, Supervision, Writing - review and editing

### Author ORCIDs

Yo Sasaki (iD) http://orcid.org/0000-0003-0024-0031
Aaron DiAntonio (iD) http://orcid.org/0000-0002-7262-0968

### Decision letter and Author response

Decision letter https://doi.org/10.7554/eLife.71148.sa1
Author response https://doi.org/10.7554/eLife.71148.sa2

## Additional files

### Supplementary files

- Transparent reporting form
- Source data 1. Data for figures.

### Data availability

All data generated or analyzed during this study are included in the manuscript and supporting files.

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
