## [Editor Report]

The authors have responded to all essential criticisms, supporting the elucidation of a sequence of cellular events that reside downstream of SARM1 activation.

---

## [Decision Letter]

**Decision letter after peer review:**

Thank you for submitting your article "Live Imaging Reveals the Cellular Events Downstream of SARM1 Activation" for consideration by *eLife*. Your article has been reviewed by 3 peer reviewers, one of whom is a member of our Board of Reviewing Editors, and the evaluation has been overseen by Suzanne Pfeffer as the Senior Editor. The reviewers have opted to remain anonymous.

Essential revisions:

1. A central point of argument in this paper concerns the relative changes in calcium influx and how these changes, when they occur, influence the sequence of events leading to mitochondria stalling, lipid exposure and axon catastrophe. The authors appropriately acknowledge that there have been a few studies assessing this issue in vivo and in vitro. But, given this, any apparent differences with the published literature should be resolved, likely with a higher level of imaging that approaches the timescale on which cell biological signaling occurs. This likely means addressing the calcium signals that occur on cell biologically relevant time scales of milliseconds to minutes. In this paper, calcium imaging is based on static images taken at intervals of minutes to hours in most cases. This seems the case for Fluo-4 in Figure one, and also for GCamp6 data. The authors argue that their data differs from published information about the timing and importance of calcium influx. But, it is more than likely that they are missing the most important calcium signaling events. The authors have not attempted to imaging signaling dynamics of calcium on fast time scales. And, when using GCamp6, it is likely that they are missing even potential slow changes in basal calcium in all but the most extreme cases. The authors are referred to : Lock et al., 2015 in which it is demonstrated that GCamp6 cannot resolve shifts in baseline calcium or calcium transients seen with other, faster, calcium indicators. The importance of this topic is emphasized by a central conclusion made by the authors on lines 283 and 284, "Calcium influx cannot explain the loss of mitochondrial mobility, so we considered other potential mechanisms". The authors may argue that degeneration occurs over a time frame of hours and this precludes fast calcium imaging. However, I see no reason that they authors could not resort to intermittent high-speed sampling of calcium signaling. This might require intermittent sampling over different time windows in different cells to cover the entire 4hr time between axon cut and degeneration.

2. The authors have focused on metabolic activity of axons by measuring mitochondria voltage and ATP levels. Given that the authors are working in cell culture, shouldn't they take advantage of this system to directly image membrane voltage and assess the integrity of the axonal membrane voltage? This would be a direct assessment of axonal health and an advance beyond what has already by pursued in other systems. It is understood that this would be a novel excursion for the authors, but this would add a new dimension not previously documented and could help define the source of changes in axonal calcium.

3. How do the authors define mitochondrial mobility? They show the total number of mitochondria that are moving in a given area, but do not specify if they are moving anterogradely or retrogradely, nor the fraction of mitochondria that are moving versus stationary. Mitochondrial movement varies widely between axon type, culture condition, and even time in culture. This speaks to whether comparing other events like mitochondrial membrane depolarization or ATP drop to mitochondrial mobility changes is a valid measure.

4. The authors use the PercevalHR genetically encoded reporter to compare the percentage drop in ATP with the percentage drop of motile mitochondria (see comment #3 above), but do not calibrate this sensor. In their previous work (Summers et al. 2020, Mol. Neurobiol.) they show that inhibition of glycolysis or oxphos can both cause a ~70% reduction in total ATP levels in these neurons in the absence of overt degeneration. That these axons have an ATP 'buffer' speaks to the need to define the relative range of ATP changes that Perceval is sensitive to in axons using pharmacological methods.

5. Left uncited are two highly relevant studies showing calcium influx prior to membrane rupture in injured axons (Yong..Deppmann, Sci.Rep 2020), and terminal exposure of phosphatidylserine on the surface of injured axons (Shacham-Silverberg…Yaron, Cell Death Disease, 2018). Indeed, the Results section should also provide more detail regarding published work from Vargas, Villegas, and Witte, currently cited. By doing a more precise job, the authors can precisely state the nature of their advance and make this clear to the reader.

6. Figure 2: To directly assess the effects of BAPTA/EGTA on axonal calcium levels it would be helpful to conduct the Fluo-4 experiments with BAPTA and EGTA treatments to show how efficient they are in suppressing calcium rise. More discussion of the different results from EGTA and BAPTA treatments would also be helpful. If EGTA blocks axon degeneration by reducing calcium influx (and therefore intracellular calcium), why does BAPTA has no effect at all, given it chelates elevated intracellular calcium too.

7. It would also help to include a Sarm1 KO dataset in for the ATP/mito analysis in Figure 5. Also, more information should be provided about how this sensor was imaged and analyzed--was the ration of fluoresence measured?

8. The authors should define "n" in all figure legends – as written it is unclear whether n represents individual axons, separate cultures, etc.*Reviewer #1:*

This paper addresses the sequence of cellular events that occurs downstream of axon severing and prior to axonal destruction. The authors emphasize the power of their in vitro assay system in which they can optically injure single axons and follow the consequences over time. The authors image ATP use, calcium levels, mitochondria mobility, mitochondria health (voltage) and membrane lipid content. Based on their imaging data, they propose a sequence of cell biological events leading to axonal destruction.

*Reviewer #2:*

Loss of Sarm1 strongly protects injured axons from metabolic collapse and morphological fragmentation. The finding that Sarm1 is an injury-induced NADase has led to a working model in the field that Sarm1 drives down NAD levels, resulting in ATP loss followed by axon fragmentation. Ko, Milbrandt, and DiAntonio seek to define and order the cellular events between Sarm1 activation and axon fragmentation and to determine which require the NADase function of Sarm1. This latter point becomes more important in light of recent work by Marc Freeman's group showing an NADase-independent role for Sarm1 in 'bystander' un-injured neurons.

The premise of this work is that many of the terminal events in Wallerian degeneration have been identified in other studies (drop in ATP, pausing of mitochondrial movement, mitochondrial membrane depolarization, calcium influx, and phosphatidylserine exposure), but that their relative order has not been established in a single experimental system. The authors rightly point out that a clear ordering of the events in Wallerian degeneration is critical, and the live imaging approach that they develop is well-suited to answer the questions they pose.

The most pressing issue is whether the findings in this study are sufficiently novel as a stand-alone study. As the authors point out, the majority of their findings have been published in other studies, and in some cases have been ordered with respect to each other. As an example, injury-induced calcium influx has been shown to immediately precede axon fragmentation, which is itself caused by calcium-dependent calpains (Vargas…Sagasti, 2015; Yang…Tessier-Lavigne, 2013; both studies are cited in this manuscript).*Reviewer #3:*

This study investigates cellular events downstream of axotomy and SARM1 activation that leads to degeneration with single axon resolution. The authors find that SARM1 NADase activity leads to an ordered sequence of events beginning with loss of ATP, followed by mitochondrial dysfunction, calcium influx, externalization of phosphatidylserine and ending with loss of membrane integrity. Importantly, the findings highlight ATP loss as the point of no return for axotomized axons.

---

## [Author Response]

Essential revisions:1. A central point of argument in this paper concerns the relative changes in calcium influx and how these changes, when they occur, influence the sequence of events leading to mitochondria stalling, lipid exposure and axon catastrophe. The authors appropriately acknowledge that there have been a few studies assessing this issue in vivo and in vitro. But, given this, any apparent differences with the published literature should be resolved, likely with a higher level of imaging that approaches the timescale on which cell biological signaling occurs. This likely means addressing the calcium signals that occur on cell biologically relevant time scales of milliseconds to minutes. In this paper, calcium imaging is based on static images taken at intervals of minutes to hours in most cases. This seems the case for Fluo-4 in Figure one, and also for GCamp6 data. The authors argue that their data differs from published information about the timing and importance of calcium influx. But, it is more than likely that they are missing the most important calcium signaling events. The authors have not attempted to imaging signaling dynamics of calcium on fast time scales. And, when using GCamp6, it is likely that they are missing even potential slow changes in basal calcium in all but the most extreme cases. The authors are referred to : Lock et al., 2015 in which it is demonstrated that GCamp6 cannot resolve shifts in baseline calcium or calcium transients seen with other, faster, calcium indicators. The importance of this topic is emphasized by a central conclusion made by the authors on lines 283 and 284, "Calcium influx cannot explain the loss of mitochondrial mobility, so we considered other potential mechanisms". The authors may argue that degeneration occurs over a time frame of hours and this precludes fast calcium imaging. However, I see no reason that they authors could not resort to intermittent high-speed sampling of calcium signaling. This might require intermittent sampling over different time windows in different cells to cover the entire 4hr time between axon cut and degeneration.

We take issue with the premise of the concern. We do report any contradictions with prior studies; we report the same basic findings as prior studies such as a late wave of calcium influx precedes axon degeneration. However, we extended these findings in ways never done before, using single axon live imaging to define the temporal relationships among this wave of calcium influx, axon degeneration, and other cell biological events that occur in the injured axon. In addition, we confirm that blocking this late wave of calcium influx inhibits morphological breakdown of axons, but for the first time assess the health of such axons and show that this treatment triggers mitochondrial depolarization and blocks mitochondrial movement. These are new findings, not disagreements with prior work. As such, we don’t see the problem that this suggestion is trying to solve.

Imaging calcium at high temporal resolution in injured axons might finding something new, but that is not the question we are addressing with this paper. For decades it has been known that there is a large and late calcium spike prior to axon degeneration. Here we provide the first quantitative analysis of the relationship between this spike in calcium and a series of cell biological changes occurring in an injured axon. This is the question we set out to address. While potentially interesting, imaging at time frames that are not obviously relevant to the process of axon degeneration is, in our opinion, a large effort that does not test a specific hypothesis.

Even if such a study would provide useful information, the suggestion of doing high speed calcium imaging is not practical in this system. The reviewers suggested high-speed calcium imaging “during the 4 hr between axon cut and degeneration.” However, as shown in figure 3, the time between axon cut and degeneration is highly variable between axons, and can occur from 3 to 9 hours after cut. The axons are less than 1 micron in width and repeated high-speed imaging, even if intermittent, will result in dramatic photobleaching over this protracted time frame.

The reviewers pointed out one sentence in our paper that they found overstated our findings, e.g. "Calcium influx cannot explain the loss of mitochondrial mobility, so we considered other potential mechanisms". We are happy to clarify this sentence to make clear that we are referring to the large and late influx of calcium that occurs prior to axon degeneration, which certainly cannot explain the loss of mitochondrial mobility since it occurs after mitochondria stop moving (a novel finding reported here for the first time). We will revise that sentence and make clear throughout the text the we are characterizing this previously studied late wave of calcium, and not drawing conclusions about calcium fluctuations that may or may not occur during axon degeneration.

2. The authors have focused on metabolic activity of axons by measuring mitochondria voltage and ATP levels. Given that the authors are working in cell culture, shouldn't they take advantage of this system to directly image membrane voltage and assess the integrity of the axonal membrane voltage? This would be a direct assessment of axonal health and an advance beyond what has already by pursued in other systems. It is understood that this would be a novel excursion for the authors, but this would add a new dimension not previously documented and could help define the source of changes in axonal calcium.

We agree that this is a good suggestion and have tried multiple voltage sensors in this system but have not succeeded in generating usable data. We think that is likely due to these axons being so thin. As the reviewers point out, this would be a novel technique to develop and while it might provide additional information, we would note that it is not necessary to support any of our current conclusions and hence is beyond the scope of this study.

3. How do the authors define mitochondrial mobility? They show the total number of mitochondria that are moving in a given area, but do not specify if they are moving anterogradely or retrogradely, nor the fraction of mitochondria that are moving versus stationary. Mitochondrial movement varies widely between axon type, culture condition, and even time in culture. This speaks to whether comparing other events like mitochondrial membrane depolarization or ATP drop to mitochondrial mobility changes is a valid measure.

We are happy to describe our methods more carefully so that it is clear how we defined mitochondrial mobility. While we agree that mobility can vary; we addressed this in figure 4B and 4C by assessing mitochondrial mobility across the entire scope of the experiment in every axon. There is some modest variability (note variability in baseline from 0-3 hrs in fig 4B), but we focused on the dramatic change when mitochondria have completely stopped moving. Even demanding such an all or none response, we still found, and quantified, that this occurred prior to the large influx of calcium that is characteristic of axon degeneration (Fig 4C). In figure 5 the change in mobility vs ATP changes was not all or none, which we freely acknowledge and quantify all of the data that lead to our measured conclusions. We did not quantify anterograde vs retrograde separately. Our observations suggested that both changed very similarly, but since we were focused on the all-or-none response of mitochondria stopping, we did not see the value in quantifying those two separately. Finally, the reviewer was concerned that we did not quantify the fraction of mitochondria moving vs stationery, but we did report the % moving. In the revised text we will make clear that 100% - % moving = % stationery.

4. The authors use the PercevalHR genetically encoded reporter to compare the percentage drop in ATP with the percentage drop of motile mitochondria (see comment #3 above), but do not calibrate this sensor. In their previous work (Summers et al. 2020, Mol. Neurobiol.) they show that inhibition of glycolysis or oxphos can both cause a ~70% reduction in total ATP levels in these neurons in the absence of overt degeneration. That these axons have an ATP 'buffer' speaks to the need to define the relative range of ATP changes that Perceval is sensitive to in axons using pharmacological methods.

As requested, we used CCCP to lower ATP and measured the effect on Perceval. In our prior studies we showed that after 2 hours 50 μM CCCP lowers ATP by ~60% in DRG neurons (Summers, 2014, Figure 1). We have now made the same manipulation and quantified Perceval intensity. Following 2 hours of 50 μM CCCP Perceval intensity drops ~70% in axons. This is now included as Figure 5-figure supplement 1.

5. Left uncited are two highly relevant studies showing calcium influx prior to membrane rupture in injured axons (Yong..Deppmann, Sci.Rep 2020), and terminal exposure of phosphatidylserine on the surface of injured axons (Shacham-Silverberg…Yaron, Cell Death Disease, 2018). Indeed, the Results section should also provide more detail regarding published work from Vargas, Villegas, and Witte, currently cited. By doing a more precise job, the authors can precisely state the nature of their advance and make this clear to the reader.

We have now cited the two relevant studies and provided a more comprehensive discussion of the contribution of prior work.

6. Figure 2: To directly assess the effects of BAPTA/EGTA on axonal calcium levels it would be helpful to conduct the Fluo-4 experiments with BAPTA and EGTA treatments to show how efficient they are in suppressing calcium rise. More discussion of the different results from EGTA and BAPTA treatments would also be helpful. If EGTA blocks axon degeneration by reducing calcium influx (and therefore intracellular calcium), why does BAPTA has no effect at all, given it chelates elevated intracellular calcium too.

We have performed the requested experiments and they are presented in a new Figure 2-figure supplement 1C and 1D. We find that EGTA efficiently blocks the rise in calcium after axotomy, while BAPTA partially reduces this rise. This is consistent with the large rise in calcium being due to influx of extracellular calcium, and supports the model that it is this extracellular calcium that promotes fragmentation of the axon. As requested we have added more discussion of these findings.

7. It would also help to include a Sarm1 KO dataset in for the ATP/mito analysis in Figure 5. Also, more information should be provided about how this sensor was imaged and analyzed--was the ration of fluoresence measured?

We have added additional details to the methods.

8. The authors should define "n" in all figure legends – as written it is unclear whether n represents individual axons, separate cultures, etc.

We now define n in the figure legends.